# Efficacy of Contrast-Enhanced Endoscopic Ultrasonography for the Differentiation of Non-Hodgkin’s Lymphoma: A Single-Center Retrospective Cohort Study

**DOI:** 10.3390/jcm12052054

**Published:** 2023-03-05

**Authors:** Kensaku Yoshida, Takuji Iwashita, Naoki Mita, Yuhei Iwasa, Shinya Uemura, Masahito Shimizu

**Affiliations:** First Department of Internal Medicine, Gifu University Hospital, Gifu 501-1194, Japan

**Keywords:** contrast-enhanced endoscopic ultrasound, non-Hodgkin’s lymphoma, time–intensity curve analysis

## Abstract

Background: Contrast-enhanced endoscopic ultrasound (CE-EUS) is a promising diagnostic modality for differentiating malignant and benign lymph nodes. This study aimed to evaluate the diagnostic capability of CE-EUS in differentiating indolent non-Hodgkin’s lymphoma (NHL) from aggressive NHL. Methods: Patients who underwent CE-EUS and endoscopic ultrasound–guided fine needle aspiration (EUS-FNA) for lymphadenopathy and were diagnosed with NHL were included in this study. Echo features on B-mode endoscopic ultrasound (EUS) and vascular and enhancement patterns on CE-EUS were qualitatively evaluated. The enhancement intensity of the lymphadenopathy on CE-EUS over 60 s was also quantitatively evaluated using time–intensity curve (TIC) analysis. Results: A total of 62 patients who were diagnosed with NHL were enrolled in this study. Regarding qualitative evaluation using B-mode EUS, there were no significant differences in the echo features between aggressive NHL and indolent NHL. With regard to qualitative evaluation using CE-EUS, aggressive NHL showed a heterogeneous enhancement pattern that is significantly more frequent than indolent NHL (95% confidence interval: 0.57 to 0.79, *p* = 0.0089). When heterogeneous enhancement was defined as aggressive NHL, the sensitivity, specificity, and accuracy of the qualitative evaluation when using CE-EUS were 61%, 72%, and 66%, respectively. In TIC analysis, the velocity of reduction for homogeneous lesions was significantly higher in aggressive NHL than in indolent NHL (*p* < 0.0001). The sensitivity, specificity, and accuracy of CE-EUS in differentiating indolent NHL from aggressive NHL improved to 94%, 69%, and 82%, respectively, when combined with qualitative and quantitative evaluations. Conclusions: CE-EUS before EUS-FNA for mediastinal or abdominal lymphadenopathy may be useful for improving the diagnostic capability of differentiating between indolent NHL and aggressive NHL (clinical trial registration number: UMIN000047907).

## 1. Introduction

Malignant lymphoma is one of the most important diseases for the differential diagnosis of lymphadenopathy in the mediastinal or abdominal cavity. non-Hodgkin’s lymphoma (NHL) is the most common malignant lymphoma and is further divided into three major categories in terms of biological behavior: indolent lymphoma, aggressive lymphoma, and highly aggressive lymphoma. The treatment strategy for NHL is generally similar to the entities within each group, although the outcomes of the individual entities within each group can vary significantly [1]. Therefore, the categorization of NHL diagnosis by using lymphadenopathy is important for determining treatment strategies. For thoracic or abdominal lymphadenopathy, the European Society of Gastrointestinal Endoscopy recommends endoscopic ultrasound-guided fine needle aspiration (EUS-FNA) for the diagnosis and subclassification of malignant lymphoma [2]. However, malignant lymphoma may be difficult to diagnose in lesions where EUS-FNA cannot be performed.

Studies have reported that contrast-enhanced endoscopic ultrasound (CE-EUS) could be a promising diagnostic modality for pancreatic tumors and lymph nodes [3,4,5,6,7,8,9,10]. However, according to these studies, various enhancement patterns have been recognized in malignant lymphoma, thus making it difficult to differentiate malignant lymphoma from other diseases with enhancement patterns by using qualitative evaluation, such as CE-EUS. The various enhancement patterns shown by malignant lymphoma in CE-EUS might depend on biological behavior. However, no study has evaluated the correlation between the enhancement pattern and biological behavior of malignant lymphoma in CE-EUS. We have previously reported that the diagnostic accuracy of CE-EUS with qualitative and quantitative analyses for differentiating malignant from benign lymphadenopathy was significantly better than the qualitative assessment of CE-EUS [11]. Thus, this study was conducted to evaluate the diagnostic capability of the qualitative and quantitative evaluations of CE-EUS in differentiating biological behavior in terms of indolent NHL versus aggressive NHL.

## 2. Materials and Methods

### 2.1. Patient Selection

This retrospective cohort study was conducted at a single center—namely, Gifu University Hospital (Gifu, Japan)—between September 2016 and August 2019. The inclusion criteria were patients who underwent CE-EUS and EUS-FNA for lymphadenopathy in the mediastinum or abdominal cavity and were diagnosed with NHL on the basis of the pathological findings obtained by EUS-FNA. However, patients with a major axis of less than 10 mm on endoscopic ultrasound (EUS) were excluded because it was difficult to determine the contrast pattern. The classification of lymphoma was based on the World Health Organization classification of lymphoid neoplasms [12], and the categories of NHL were based on the World Health Organization classification of lymphomas [13]. The study was performed in accordance with the Declaration of Helsinki. The study protocol was approved by the institutional review board of Gifu University Hospital, and this study was registered in the UMIN Clinical Trials Registry (UMIN000047907).

### 2.2. EUS Procedure

All patients were moderately sedated with midazolam and pentazocine during the EUS procedure. EUS was performed with a convex-type echoendoscope (GF-UCT260; Olympus, Tokyo, Japan) connected with US systems (ProSound F75 or Pro-Sound alpha 10; FUJIFILM, Tokyo, Japan). Before CE-EUS, the size, shape (round or oval), border (sharp or fuzzy), echogenicity (hyperechoic or hypoechoic), and echo texture (heterogeneous or homogeneous) of each lymph node was evaluated. With regard to CE-EUS, the mechanical index was set at 0.2, and the transmitting frequency was set at 5 MHz, respectively. The extended pure harmonic detection mode was used for CE-EUS. Sonazoid^®^ (Daiichi-Sankyo, Tokyo, Japan, or GE Healthcare, Milwaukee, WI, USA) was injected as the contrast agent for CE-EUS. A measure of 0.015 mL/kg body weight of the contrast agent was injected through a peripheral vein by bolus injection. After injection of the contrast enhancer, the images acquired by CE-EUS were recorded for 90 s. Digital Imaging and Communications in Medicine (DICOM) format and Audio Video Interleaved (AVI) format were used for recording all the examinations. EUS-FNA was performed for all lymph nodes, followed by CE-EUS. Patients were observed in the endoscopy unit for at least 2 h after the procedure.

### 2.3. Image Analysis

As for image analysis of CE-EUS, the enhancement patterns were categorized as homogeneous or heterogeneous, and the vascular patterns were classified as hypervascular or hypovascular, respectively. Vascular and enhancement patterns were evaluated at 30–45 s. We previously reported that vascular patterns of hypervascular and isovascular are difficult to distinguish in a lymph node because of the connective tissue around the organ; therefore, vascular patterns characterized as hypervascular and isovascular were all considered hypervascular [11]. Similar to the previous study [11], hypervascular lesions were defined as being similar to higher intensity of enhancement compared to the surrounding connective tissue; on the other hand, hypovascular lesions were defined as a lower intensity of enhancement compared to the surrounding connective tissue in this study. 

### 2.4. Time Intensity Curve (TIC) Analysis

All the examinations of CE-EUS were recorded in DICOM format, and TIC analysis was performed using DAS-RS1 (Hitachi-Aloka Medical, Tokyo, Japan). For the TIC analysis, the average echo intensity within the region of interest was used. The region of interest was set as to trace the outline of the lesion at the time of administration of the contrast medium. TIC analysis was performed for up to 60 s after the administration of the contrast medium. The baseline intensity (I_base_, dB) was defined as intensity before administration of contrast medium. The intensity at 60 s was defined as I_60_ (dB). Maximum intensity was I_max_ (dB); maximum echo intensity was from I_base_ to I_60_, time to maximum intensity was T_max_ (seconds); intensity gain from time from I_max_ to I_0_ was I_max_ − I_base_ (dB); intensity gain from I_base_ to I_max_ and velocity of reduction (VR) was [I_max_ − I_60_]/[60 − T_max_], (dB/s); and VR from I_max_ to I_60_ was evaluated by TIC analysis (Figure 1). TIC analysis was carried out separately for lesions with homogeneous and heterogeneous enhancement by CE-EUS in the current study, similar to that in a previous study [11].

### 2.5. Statistical Analysis

In order to assess the efficacy of qualitative evaluation in CE-EUS, the sensitivity, specificity, positive predictive value (PPV), negative predictive value (NPV), and accuracy were calculated. For the comparison between the two groups, either Fisher’s exact test or the chi-square test was used for nominal variables. The Mann–Whitney U test was used for continuous variables. To determine sensitivity, specificity and accuracy with the optimal cut-off value, we performed Receiver operating characteristic (ROC) analysis on TIC parameters with statistical significance. A *p*-value < 0.05 was considered statistically significant. All statistical analyses were carried out with JMP software version 11.2 (SAS Institute, Cary, NC, USA).

## 3. Results

### 3.1. Baseline Patient Characteristics

A total of 62 patients (36 male and 26 female) out of 188 patients who underwent EUS-FNA for mediastinal or abdominal lymphadenopathy around the upper gastrointestinal tract detected by computed tomography (CT), magnetic resonance imaging, or positron emission tomography (PET)-CT and diagnosed NHL were enrolled during the study period (Figure 2). The median age of the patients was 69 years (range 31–86). Four out of 62 lesions were mediastinal, and 58 out of 62 lesions were abdominal. The median follow-up period was 742 days (range 244–1322). The final diagnoses were aggressive NHL in 33 patients (diffuse large B-cell lymphoma in 27 patients, peripheral T-cell lymphoma in two patients, extranodal Natural Killer (NK)/T-cell lymphoma in 1 patient, plasmablastic lymphoma in one patient, T-cell lymphoma/histiocyte-rich B-cell lymphoma in one patient, and adult T-cell leukemia/lymphoma in one patient) and indolent NHL in 29 patients (follicular lymphoma in 27 patients, mantle cell lymphoma in one patient, and mucosa-associated lymphoid tissue (MALT) lymphoma in one patient). The baseline characteristics of study patients are presented in Table 1.

### 3.2. Qualitative Evaluation in B-Mode EUS

The median long axes of the region for the indolent NHLs and aggressive NHLs were 23 mm and 29 mm, respectively, and the median short axes for the indolent NHLs and aggressive NHLs were 17 mm and 21 mm, respectively. The shape of the indolent NHLs and aggressive NHLs was round in 17 patients and 17 patients, respectively, and oval in 12 patients and 16 patients, respectively. The borders of the indolent NHLs and aggressive NHLs were sharp in 18 patients and 15 patients, respectively, and fuzzy in 11 and 18 patients, respectively. The echogenicity of the aggressive NHLs and indolent NHLs was hyperechoic in two patients and one patient, and hypoechoic in thirty-one patients and twenty-eight patients, respectively. The echotexture of the aggressive NHLs and indolent NHLs was heterogeneous in 21 patients and 14 patients, and homogeneous in 12 patients and 15 patients, respectively (Table 2). For qualitative evaluation in B-mode EUS, there were no significant differences between indolent NHL and aggressive NHL in terms of the long axis, short axis, shape, border, echogenicity, and echotexture.

### 3.3. Qualitative Evaluation in CE-EUS

The enhancement and vascular patterns were divided into three patterns: homogeneous enhancement and hypervascular in 13 patients with aggressive NHL and 21 patients with indolent NHL (Figure 3a); heterogeneous enhancement and hypervascular in five patients with aggressive NHL (Figure 3b); and heterogeneous enhancement and hypovascular in fifteen patients with aggressive NHL and eight patients with indolent NHL (Figure 3c). The enhancement and vascular patterns of 62 patients in CE-EUS are presented in Table 3. Aggressive NHLs often exhibited heterogeneous enhancement that were statistically significant compared with indolent NHLs (*p* = 0.0089). When heterogeneous enhancement was defined as an aggressive NHL, the sensitivity, specificity, PPV, NPV, and accuracy of the qualitative evaluation using CE-EUS were 61%, 72%, 71%, 62%, and 66%, respectively. No adverse events related to CE-EUS were observed in this study.

### 3.4. Quantitative Evaluation in TIC 

Table 4 presents the results of the TIC analysis. The TIC analysis showed no significant difference between indolent and aggressive NHL patients in heterogeneous enhancement. However, TIC analysis showed a significant difference in homogeneous enhancement. The VR was significantly faster in aggressive NHL compared to indolent NHL (*p* < 0.0001) (Figure 4). The ROC analysis of VR in homogeneous enhancement for indolent NHLs showed an area under the curve of 0.92 and a cut-off value of 0.16. dB/s (Figure 5). At this cut-off value, the sensitivity, specificity, and accuracy of the TIC analysis of aggressive NHLs of lesions with homogeneous enhancement were 86%, 95%, and 91%, respectively. The sensitivity, specificity, PPV, NPV, and accuracy of CE-EUS for aggressive NHL improved to 94%, 69%, 78%, 91%, and 82%, respectively, by combining qualitative and quantitative evaluations. The results of qualitative and quantitative evaluations in CE-EUS are presented in Table 5. CE-EUS combining qualitative and quantitative evaluations in differentiating aggressive NHL from indolent NHL resulted in a significantly higher accuracy rate than the qualitative evaluation of CE-EUS. (Table 6).

## 4. Discussion

The diagnostic ability of the qualitative evaluation of CE-EUS for differentiating indolent NHL and aggressive NHL in mediastinal or abdominal lymphadenopathy showed a sensitivity, specificity, and accuracy of 61%, 72%, and 66%, respectively. Quantitative evaluation with TIC analysis improved the sensitivity, specificity, and accuracy to 94%, 69%, and 82%, respectively, when combined with qualitative and quantitative evaluations. The accuracy rate of CE-EUS combining qualitative and quantitative evaluations in differentiating indolent NHL and aggressive NHL was significantly higher than that of the qualitative evaluation of CE-EUS (*p* = 0.0389).

Several studies have reported the differentiation of NHL using sonographic features. Gu et al. [14] evaluated the B-mode sonographic characteristics of thyroid ultrasound for primary thyroid lymphoma (PTL) in 27 patients. They reported in the discussion section that of 25 PTLs with echogenic strands, all (8/8) MALTs and 73.3% (11/15) of DLBCLs had linear echogenic strands, and two DLBCLs had destructive linear strands; therefore, it is difficult to estimate the pathological type of lymphoma on the basis of sonographic characteristics. Javier et al. [15] evaluated the sonographic findings of transabdominal ultrasound for the liver involvement of histologically proven NHL with histologic subtype in 17 patients. They reported that there were no differences in sonographic patterns (hepatomegaly, splenomegaly, focal liver lesions, diameter, echogenicity, and lymphadenopathies) between aggressive NHL in thirteen patients and indolent NHL in four patients. In the current study, there were no significant differences in sonographic characteristics (long axis, short axis, shape, border, echogenicity, and echotexture) from B-mode EUS between aggressive NHLs and indolent NHLs. It seems to be difficult to distinguish between aggressive NHL and indolent NHL by using only sonographic characteristics based on B-mode EUS.

There have been two reports on the usefulness of contrast-enhanced ultrasound (CE-US) in differentiating between indolent and aggressive NHL. Jing et al. [16] reported 76 lymph nodes of NHL, including indolent NHL in 14 lymph nodes and aggressive NHL in 62 lymph nodes, with respect to the value of CE-US. They reported that 1 out of 14 cases of indolent NHL and 10 out of 62 cases of aggressive NHL showed heterogeneous enhancement. There was no significant difference between indolent NHL and aggressive NHL in terms of enhancement pattern (*p* = 0.3540); however, aggressive NHL tended to show heterogeneous enhancement compared with indolent NHL. Xuelei et al. [17] reported 140 lymphomatous lymph nodes, including aggressive NHL in 136 lymph nodes and indolent NHL in four lymph nodes, concerning the application of CE-US. They reported that 81.6% of aggressive NHLs and 100% of indolent NHLs showed rapid well-distributed hyperenhancement and that 18.4% of aggressive NHLs showed rapid heterogeneous hyperenhancement. In this study, rapid well-distributed hyperenhancement indicated hypervascularity with homogeneous enhancement, and rapid heterogeneous hyperenhancement indicated hypervascularity with heterogeneous enhancement. The two reports indicate that many NHLs show homogeneous enhancement in CE-US, and it is difficult to distinguish between indolent NHL and aggressive NHL in CE-US with qualitative assessment alone. In the present study, 63% of aggressive NHL cases and 37% of indolent NHL cases showed heterogeneous enhancement, with a statistically significant difference. Our study results showed that aggressive NHL might show more heterogeneous enhancement, although it is difficult to distinguish between aggressive NHL and indolent NHL by using only the qualitative evaluation of CE-EUS for mediastinal or abdominal lymphadenopathy.

In this study, TIC analysis was performed on lesions showing homogeneous and heterogeneous enhancements in the qualitative evaluation. No significant difference was found in heterogeneous enhancement. However, aggressive NHL resulted in significantly faster VR than indolent NHL in lesions with homogeneous enhancement (*p* < 0.0001). Jiang et al. [16] reported on the usefulness of quantitative evaluation with TIC analysis of CE-US for the differentiation of indolent and aggressive NHL. They performed CE-US on lymph nodes for 90 s, set the region of interest around the margin of lesion in the lymph node, and calculated the maximum intensity, time to maximum intensity, and VR, similar to the current study. There were no statistical differences in the TIC parameters, including maximum intensity, time to maximum intensity, and VR, in the identification of indolent NHL and aggressive NHL; however, the VR showed a faster tendency in aggressive NHL than in indolent NHL (*p* = 0.088). They speculated that lesions with a higher degree of aggressiveness might have richer tumor vascularity, which might increase blood velocity. This might be the reason why aggressive NHL resulted in significantly faster VR than indolent NHL in this study. 

Several studies [18,19,20] reported that aggressive NHL showed a significantly higher SUV max than that of indolent NHL with PET-CT. In this study, PET-CT was performed in 40 patients and showed that aggressive NHLs had a significantly higher SUV max than indolent NHLs (aggressive NHL vs indolent NHL: 15.12 vs 8.76; *p* = 0.0263). Based on the results of ROC analysis (AUC of 0.7133 with the cut-off level of 4.4), the diagnostic accuracy of aggressive NHL by PET-CT was 72%. CE-EUS qualitative and quantitative analysis for these 40 cases showed an accuracy rate of 90%, which was significantly higher in comparison with the results of PET-CT (*p* = 0.0416). Although it was a small number of cases, CE-EUS may be more useful than PET-CT in differentiating between indolent and aggressive NHL.

By using a combination of qualitative and quantitative evaluations in CE-EUS for mediastinal or abdominal lymphadenopathy, the current study showed that the accuracy rate for distinguishing between indolent NHL and aggressive NHL was 82%. In 13 out of 34 patients who showed as hypervascular with homogeneous enhancement via the qualitative evaluation of CE-EUS, it was difficult to distinguish between indolent NHL and aggressive NHL by using qualitative evaluation alone. However, when combining qualitative and quantitative evaluations, CE-EUS was able to distinguish between indolent NHL and aggressive NHL for 31 out of 34 patients. (VR: aggressive NHL, 0.182 dB/s; indolent NHL, 0.127 dB/s; *p* < 0.0001). Although there were some lesions in which the differentiation of aggressive NHL from indolent NHL was difficult even with the application of a combination of qualitative and quantitative evaluations, Ribeiro et al. [21] showed that indolent NHL might be difficult to diagnose even with EUS-FNA. Performing CE-EUS before EUS-FNA for mediastinal or abdominal lymphadenopathy may still assist in diagnosis in cases wherein it is difficult to determine the biological grade of malignant lymphoma.

This study has several limitations. A retrospective study design with a small cohort may have caused bias in patient selection. There was no comparison arm to evaluate CE-EUS in distinguishing indolent NHL from aggressive NHL. The most notable strength of this study is that this is the first study that distinguished indolent NHL and aggressive NHL by using CE-EUS.

## 5. Conclusions

In conclusion, the qualitative evaluation of CE-EUS for lymphadenopathy showed a certain diagnostic ability in the categorization of NHL. Moreover, the accuracy rate was further improved by examining the VR in the TIC analysis. Performing CE-EUS before EUS-FNA for mediastinal or abdominal lymphadenopathy may be useful in improving the diagnostic capability of differentiating between indolent and aggressive NHL.

## Figures and Tables

**Figure 1 jcm-12-02054-f001:**
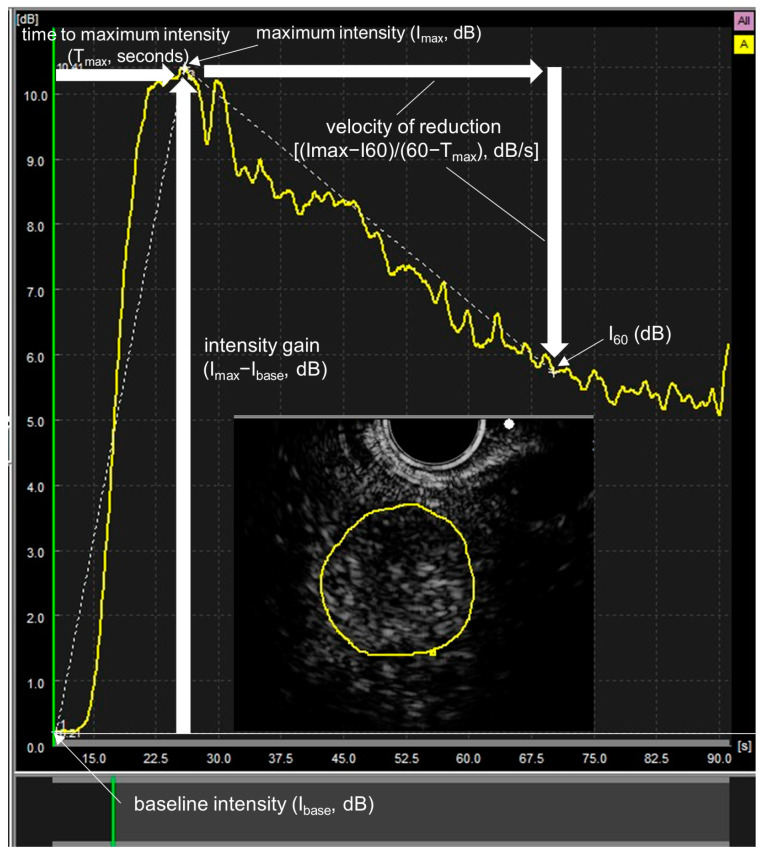
Quantitative evaluation using time–intensity curve (TIC) analysis for contrast enhanced endoscopic ultrasonography in malignant lymphoma. The region of interest is indicated by a yellow line. The following parameters were examined by TIC analysis. Baseline intensity (I_base_, dB)—echo intensity before contrast medium administration; I_60_—echo intensity at 60 s; maximum intensity (I_max_, dB)—maximum echo intensity from I_base_ to I_60_; time to maximum intensity (T_max_, s)—time to I_max_ from I; intensity gain (I_max_ − I_base_, dB)—intensity gain from I_base_ to I_max_; velocity of reduction ([I_max_ − I_60_]/[60 − T_max_], dB/s)—velocity of reduction from I_max_ to I_60_.

**Figure 2 jcm-12-02054-f002:**
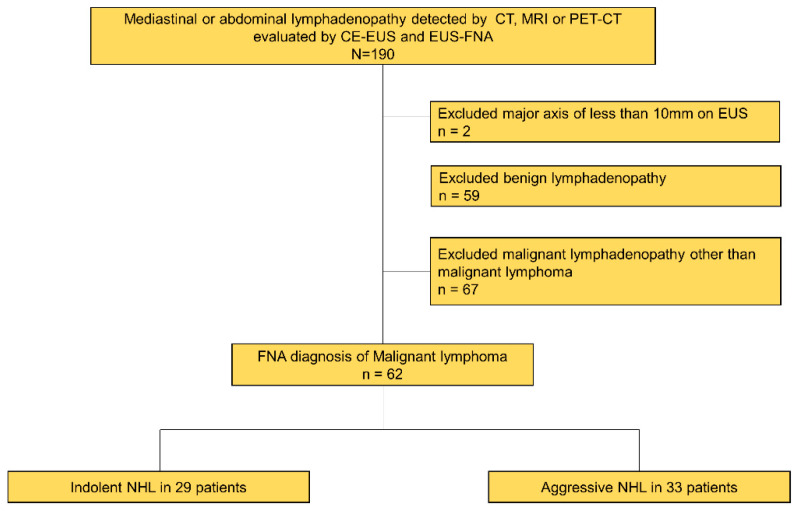
Flow chart of study enrollment. EUS: endoscopic ultrasound, CE-EUS: contrast enhanced-EUS, EUS-FNA: EUS-guided fine-needle aspiration, CT: computed tomography, MRI: magnetic resonance imaging, PET-CT: positron emission tomography-CT, NHL: non-Hodgkin’s lymphomas.

**Figure 3 jcm-12-02054-f003:**
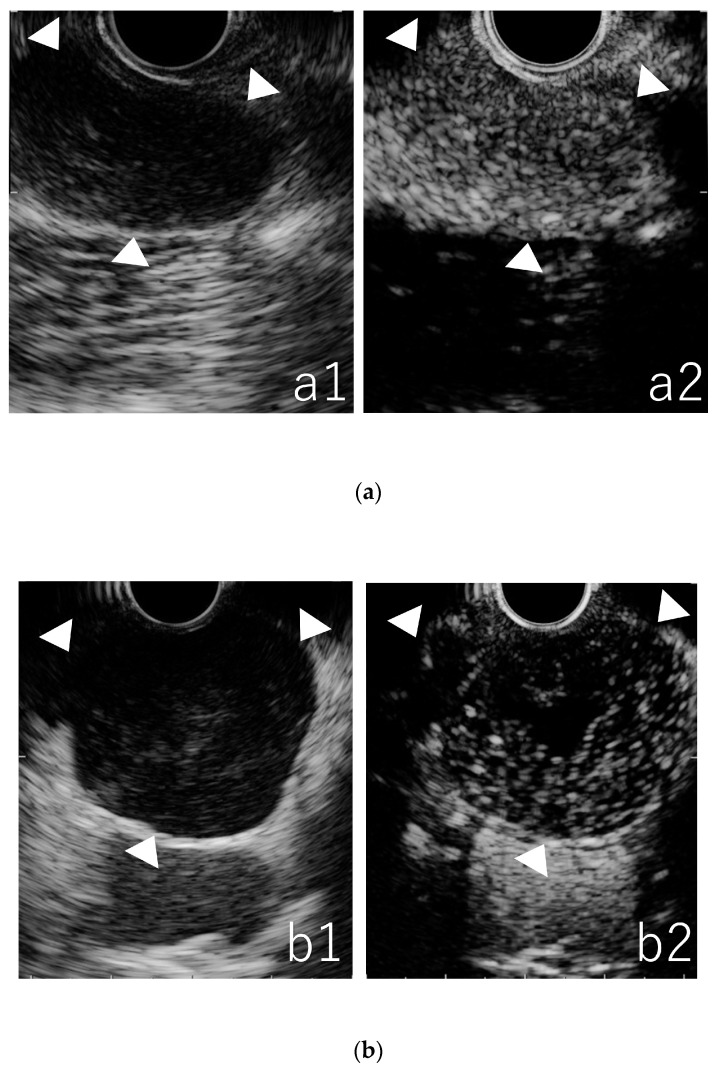
**The three patterns of figures for vascular and enhancement patterns of non-Hodgkin’s lymphoma. The white arrow shows each lesion.** (**a**) Typical lesion of a lymph node that shows homogeneous enhancement and hypervascular (an indolent non-Hodgkin’s lymphomas; Follicular lymphoma). (**a1**) fundamental B-mode, (**a2**) contrast-enhanced mode. (**b**) Typical lesion of a lymph node that shows heterogeneous enhancement and hypervascular (an aggressive non-Hodgkin’s lymphomas; diffuse large B-cell lymphoma). (**b1**) Fundamental B-mode, (**b2**) contrast-enhanced mode. (**c**) Typical lesion of a lymph node that shows heterogeneous enhancement and hypovascular (an aggressive non-Hodgkin’s lymphomas; diffuse large B-cell lymphoma) (**c1**) fundamental B-mode, (**c2**) contrast-enhanced mode.

**Figure 4 jcm-12-02054-f004:**
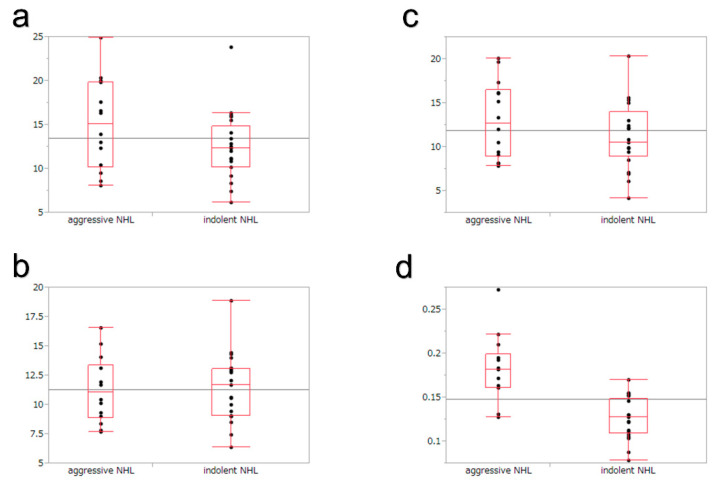
Comparison of indolent and aggressive non-Hodgkin’s lymphomas in lesions with homogeneous enhancement by time–intensity curve (TIC) analysis. TIC parameters are as follows; (**a**) time to maximum intensity, (**b**) maximum intensity, (**c**) intensity gain, (**d**) velocity of reduction.

**Figure 5 jcm-12-02054-f005:**
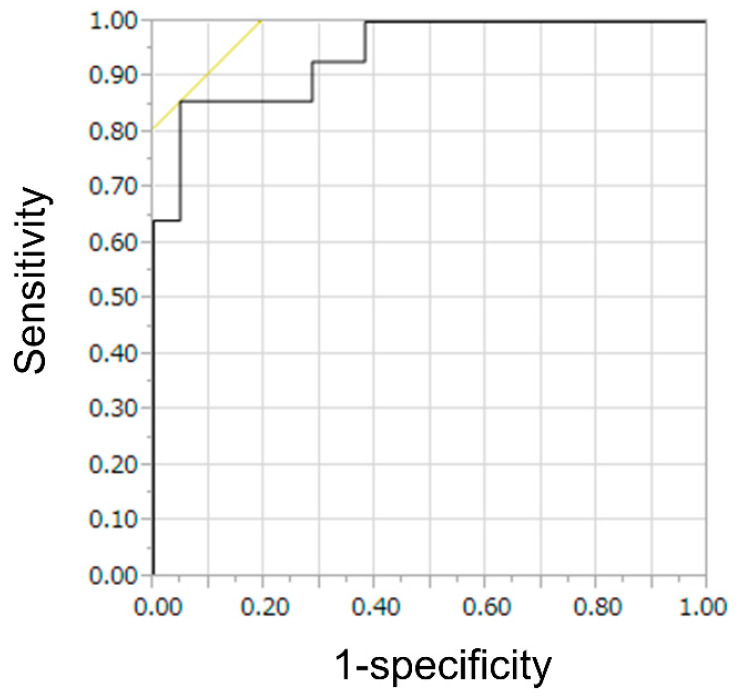
Receiver operating characteristic (ROC) analysis for velocity of reduction of time–intensity curve analysis in lesion of homogeneous enhancement. ROC analysis for velocity of reduction shows that the area under the curve is 0.92.

**Table 1 jcm-12-02054-t001:** Baseline characteristics of the patients.

		Aggressive NHL	Indolent NHL	*p* Value
**Age, year old, median (Range)**		69 (31–86)	68 (54–83)	0.9077
**Sex, n**	Male	20	16	0.6653
	Female	13	13	
**Location, n**	Mediastinum	3	1	0.3545
	Abdominal	30	28	
**Aggressive NHL, n**	Diffuse large B-cell lymphoma	27		
	Peripheral T-cell lymphoma	2		
	Extranodal NK/T-cell lymphoma	1		
	Plasmablastic lymphoma	1		
	T-cell lymphoma/histiocyte rich B-cell lymphoma	1		
	Adult T-cell leukemia/lymphoma	1		
	Total	33		
**Indolent NHL, n**	Follicular lymphoma		27	
	Mantle cell lymphoma		1	
	MALT lymphoma		1	
	Total		29	

NHL: Non-Hodgkin’s Lymphoma.

**Table 2 jcm-12-02054-t002:** The sonographic features of NHL.

		Aggressive NHL (N = 33)	Indolent NHL (N = 29)	*p* Value
**Size of region, mm, median (range)**	short axis	21 (7–75)	17 (4–85)	0.3057
	long axis	29 (10–80)	23 (10–98)	0.4629
**Shape, n**	round	17	17	0.5745
	oval	16	12	
**Border, n**	sharp	15	18	0.1895
	fuzzy	18	11	
**Echogenecity, n**	hyperechoic	2	1	0.6324
	hypoechoic	31	28	
**Echotexture, n**	heterogeneous	21	14	0.2236
	homogeneous	12	15	

NHL: Non-Hodgkin’s Lymphoma.

**Table 3 jcm-12-02054-t003:** The vascular and enhancement patterns of non-Hodgkin’s lymphoma in mediastinal or abdominal cavity.

	Hypervascular/Homogeneous	Hypervascular/Heterogeneous	Hypovascular/Heterogenous	*p* Value
Aggressive NHL, n	13	5	15	
Diffuse large B-cell lymphoma, n	10	5	12	0.5601
Peripheral T-cell lymphoma, n	1	0	1	0.754
Extranodal NK/T-cell lymphoma, n	0	0	1	0.3121
Plasmablastic lymphoma, n	1	0	0	0.1668
T-cell lymphoma/histiocyte rich B-cell lymphoma, n	0	0	1	0.3121
Adult T-cell leukemia/lymphoma, n	1	0	0	0.1668
Indolent NHL, n	21	0	8	
Follicular lymphoma, n	20	0	7	0.4855
Mantle cell lymphoma, n	0	0	1	0.1022
MALT lymphoma, n	1	0	0	0.4169

NHL: Non-Hodgkin’s Lymphoma.

**Table 4 jcm-12-02054-t004:** The results of time–intensity curve analysis.

	Homogeneous			Heterogeneous		
	Indolent NHL	Aggressive NHL	*p* Value	Indolent NHL	Aggressive NHL	*p* Value
Peak intensity, dB	12.3	13.9	0.178	4.9	5	0.3597
Time to peak, seconds	11.7	10.5	0.6577	11.7	11.1	0.8988
Intensity gain, dB	10.5	12	0.4565	4.5	4.7	0.3467
Velocity of reduction, dB/s	0.127	0.182	<0.0001	0.057	0.066	0.4014

NHL: Non-Hodgkin’s Lymphoma.

**Table 5 jcm-12-02054-t005:** Diagnoses of contrast enhanced endoscopic ultrasound with qualitative and quantitative evaluation.

	Diagnosis of CE-EUS	
	Indolent NHL	Aggressive NHL
Indolent NHL, n	20	9
Follicular lymphoma, n	19	8
Mantle cell lymphoma, n	0	1
MALT lymphoma, n	1	0
Aggressive NHL, n	2	31
Diffuse large B-cell lymphoma, n	1	26
Peripheral T-cell lymphoma, n	0	2
Extranodal NK/T-cell lymphoma, n	0	1
Plasmablastic lymphoma, n	0	1
T-cell lymphoma/histiocyte rich B-cell lymphoma, n	0	1
Adult T-cell leukemia/lymphoma, n	1	0

CE-EUS: contrast enhanced endoscopic ultrasound, NHL: non-Hodgkin’s Lymphoma.

**Table 6 jcm-12-02054-t006:** The diagnostic performance of contrast-enhanced endoscopic ultrasound with qualitative and quantitative evaluations compared with qualitative evaluation.

	Sensitivity (95% CI)	Specificity (95% CI)	PPV (95% CI)	NPV (95% CI)	Accuracy (95% CI)
CE-EUS qualitative evaluation (%)	61 (44–75)	72 (54–85)	71 (53–85)	62 (45–76)	66 (54–77)
CE-EUS combined evaluation (%)	94 (80–98)	69 (51–83)	78 (62–88)	91 (72–97)	82 (71–90) ^†^

PPV: positive predictive value; NPV: negative predictive value; CE-EUS: contrast enhanced endoscopic ultrasound; CE-EUS combined analysis: qualitative and quantitative evaluations of CE-EUS; † Accuracy of CE-EUS combined evaluation was significantly different (*p* < 0.05) from that of CE-EUS qualitative evaluation.

## Data Availability

Not applicable.

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
