# Peer review of "Efficacy of Contrast-Enhanced Endoscopic Ultrasonography for the Differentiation of Non-Hodgkin’s Lymphoma: A Single-Center Retrospective Cohort Study"

_jcm, 2023, doi:10.3390/jcm12052054_

Round 1
Reviewer 1 Report
Line 68, spelling of Introduction.
Would include how other imaging modalities differentiate aggressive vs indolent lymphomas such as pet/ct.
It is mentioned that some of the patients evaluated had a prior Pet/ct. Can you comment on how CE-EUS compares to other imaging modalities in differentiating indolent vs aggressive lymphomas, specifically pet/ct?
Can you comment on where you see this fitting into clinical practice since a diagnosis of lymphoma will still require a biopsy and pathology diagnosis regardless of whether it is indolent or more aggressive?
I think the study is interesting overall, but would be hard to use in practice unless it can reliably rule out a higher grade lymphoma. Potentially could it be combined with lab data etc.?
Author Response
Response to Reviewer 1 Comments
Comment 1: It is mentioned that some of the patients evaluated had a prior Pet/ct. Can you comment on how CE-EUS compares to other imaging modalities in differentiating indolent vs aggressive lymphomas, specifically pet/ct?
Response 1: In this study, 40 patients underwent PET-CT. Several studies reported that aggressive NHL showed significantly higher SUV max than indolent NHL in PET-CT. In this study, aggressive NHL also showed significantly higher SUV max than indolent NHL (aggressive NHL vs indolent NHL:15.12 vs 8.76)(P = .0263) for the 40 patients in PET-CT. Based on the results of ROC analysis (AUC: .7133, Cut off:14.4), the diagnostic accuracy of aggressive NHL by PET-CT was 72%. The results of CE-EUS with qualitative and quantitative analysis for these 40 cases had an accuracy rate of 90%, and compared with the results of PET-CT, the accuracy rate of CE-EUS with qualitative and quantitative analysis was significantly higher (P = .0416). Although a small number of cases, CE-EUS may be more useful than PET-CT in differentiating between indolent and aggressive NHL.
Comment 2: Can you comment on where you see this fitting into clinical practice since a diagnosis of lymphoma will still require a biopsy and pathology diagnosis regardless of whether it is indolent or more aggressive?
Response 2: Considering the higher accuracy for diagnosis of lymphoma in CE-EUS with qualitative and quantitative analysis, CE-EUS might be able to provide an appropriate target lesion for EUS-FNA in patients with multiple lymphadenopathies; however, at this moment, pathological diagnosis is necessary for decide treatment strategy.
Point 3: I think the study is interesting overall, but would be hard to use in practice unless it can reliably rule out a higher grade lymphoma. Potentially could it be combined with lab data etc.?
Response 3: Thank you for the comment. Since it is difficult to rule out higher grade lymphoma in clinical practice, it is considered necessary to combine lab data, CT, MRI, and PET-CT for evaluation with CE-EUS.
Reviewer 2 Report
Thank you kindly for the opportunity to review this manuscript. This is a single-center retrospective cohort study conducted in Japan comparing the diagnostic performance of contrast-enhanced endosonagraphy to differentiate aggressive and indolent Non-Hodgkin’s lymphoma. The manuscript is an important contribution to our current practice as CE-EUS is a novel tool and there has previously been limited published data on its ability to aid in the diagnosis of mediastinal or abdominal lymphadenopathy.
I do have some concerns regarding its methodology and conclusions. These are stated below.
EUS-FNA in its current form is a relatively safe procedure with few complications. Why do the authors think that this methodology they describe (CE-EUS) should replace FNA for confirmatory diagnosis? What do they mean when they say “it is ideal for differentiation to be performed without EUS FNA because adverse events can be caused by this procedure and because lesions cannot be punctured by EUS FNA.”?
How were the final diagnoses made for subtypes of NHL? Were all diagnoses based on the FNA specimens? FNA specimens can have limited diagnostic yield for differentiating subtypes of NHL. Incorrect diagnoses by FNA may have affected the interpretation of their data regarding the performance of CE-EUS as a diagnostic tool.
How long did these CE-EUS measurements take and how much did they add to the overall procedure time? Did patients tolerate this extended procedure time well?
How generalizable is this? What is the interrater variability of these assessments and measurements? How many different providers performed the analysis in this study?
Can the authors better describe their methods for contrast injection? Was the contrast agent infused at each lymph node station?
The following minor grammatical and typographical errors were identified and suggestions for improvement were provided.
Page 1, Line 6 – There should be a 1 next to every name.
Page 1, Line 9 – There should be a 1 before the affiliation name begins.
Page 2, Line 49 – Change “s” to “seconds.”
Page 2, Line 56 – Include confidence interval from the value (p = .0089), in case available.
Page 3, Line 68 – Change “INTORODUCTION” TO “INTRODUCTION.”
Page 3, Lines 79-81 – Add reference. This is a heavy statement which should be supported. If no reference available, please change the wording. If possible, provide examples of the types of adverse events.
Page 4, Line 123 – Change “s” to “seconds.”
Page 4, Line 127 – Change “h” to “hours.”
Page 4, Line 133 – Change “s” to “seconds.”
Page 5, Line 146 – Change “s” to “seconds.”
Page 5, Line 148 – Change “s” to “seconds.”
Page 5, Line 149 – Change “s” to “seconds.”
Page 5, Line 149 – Change “(Tpeak, s)” to “(Tpeak/seconds).”
Page 6, Line 175 – Consider adding “Natural Killer” to specify what “NK” stands for.
Page 6, Line 179 – Add the word “baseline” before “patient characteristics” for clarification.
Page 6, Line 190 – Change “echo texture” to “echotexture.”
Page 6, Lines 193-194 – Change “echo texture” to “echotexture.”
Page 7, Line 243 – This is a very specific statement. Please add supporting reference or clarify if it comes from reference 14.
Page 7, Lines 246-247 – Change “non-Hodgkin’s lymphoma” to “NHL.”
Page 8, Lines 251-252 – Change “echo texture” to “echotexture.”
Page 9, Line 285 – Change “s” to “seconds.”
Page 9, Lines 299-301 – This sentence should have been written differently, please look at line 300; “patients were able to distinguish” and change the wording of the whole sentence to clarify that the patients were not doing the distinctions themselves.
Page 13, Line 401 (footer from figure 1) – Change “second” to “seconds.”
Page 13, Line 406 – Move the word “Figure 2” to page 14.
Page 14, Line 416 (footer from figure 3a) – Change “B mode” to “B-mode.”
Page 15, Line 422 (footer from figure 3b) – Change “B mode” to “B-mode.”
Page 16, Line 428 (footer from figure 3c) – Change “B mode” to “B-mode.”
Page 18, Lines 460-472 – Blank page, please delete it.
Page 19 – Add the word “baseline” before “characteristics” for clarification.
Page 19, Table 1, Column 1 – Sex, n (%); percentage (%) is mentioned but not provided. Only “n” values are evident.
Page 19, Table 1, Column 1 – Location, n (%); percentage (%) is mentioned but not provided. Only “n” values are evident.
Page 20 – Blank page, please delete it.
Page 21, Table 2, Column 1 – The word “echo texture” should be changed to “echotexture.”
Page 21, Table 2, Column 2 – The word “shape” should be changed to “sharp.”
Page 21, Table 2, Title – For consistency, the title of the table should be moved to the top of the table.
Page 22, Table 3, Header – For consistency, the first letter of each of word found in the header of the table should be an uppercase letter.
Page 23, Table 4, Header – For consistency, the first letter of each of word found in the header of the table should be an uppercase letter.
Page 23, Table 4, Column 1 – Change “Time to peak, s” to “Time to peak, seconds.”
Page 23, Table 4, column 4 – Consider changing the number in red to black and then using bold fonts in case highlighting is necessary.
Page 24 – Blank page, please delete it.
Page 25, Table 5, Title – Consider not using abbreviations for “CE-EUS.”
Page 26, Table 6, Title – Change “characteristic” to “characteristics.”
Page 26, Table 6, Title – Consider not using abbreviations for “CE-EUS.”
Page 26, Table 6, Footer, Line 2 – After “combined analysis” change ; to :
Page 26, Table 6, Footer, Line 3 – Change “difference” to “different.”
Page 26, Table 6, Footer, Line 3 – Consider adding “Qualitative analysis” before the word “accuracy.”
Author Response
Response to Reviewer 2 Comments
Comment 1: EUS-FNA in its current form is a relatively safe procedure with few complications. Why do the authors think that this methodology they describe (CE-EUS) should replace FNA for confirmatory diagnosis? What do they mean when they say “it is ideal for differentiation to be performed without EUS FNA because adverse events can be caused by this procedure and because lesions cannot be punctured by EUS FNA.”?
Response 1: CE-EUS may be useful to complement EUS-FNA rather than a replacement for EUS-FNA. I would like to change the sentence “it is ideal for differentiation to be performed without EUS FNA because adverse events can be caused by this procedure and because lesions cannot be punctured by EUS FNA.” to “it may be difficult to diagnose malignant lymphoma in lesions where EUS-FNA cannot be performed.”
Point 2: How were the final diagnoses made for subtypes of NHL? Were all diagnoses based on the FNA specimens? FNA specimens can have limited diagnostic yield for differentiating subtypes of NHL. Incorrect diagnoses by FNA may have affected the interpretation of their data regarding the performance of CE-EUS as a diagnostic tool.
Response 2: All the final diagnoses of NHL subtypes were based on the pathological results of FNA and clinical course for more than 6 months.
Point 3: How long did these CE-EUS measurements take and how much did they add to the overall procedure time? Did patients tolerate this extended procedure time well?
Response 3: The measurements of CE-EUS took 1 minute, and it took an additional 1 minute and 30 seconds overall. Since the patients were administered a sedative, it was thought that the pains of patients were almost the same as that of normal treatment.
Point 4: How generalizable is this? What is the interrater variability of these assessments and measurements? How many different providers performed the analysis in this study?
Response 4: If the results of TIC analysis are proven in real time, there is a possibility that it can be generalized. However, it is difficult to generalize at this moment because it is necessary to analyze the obtained video separately. The images were evaluated by three people, and the matching rate was κ coefficient: 0.84.
Point 5: Can the authors better describe their methods for contrast injection? Was the contrast agent infused at each lymph node station?
Response 5: Among multiple lymph nodes, we selected the target lymph node for puncture by FNA, and administered a contrast agent while visualizing the lymph node by EUS.
Point 6: The following minor grammatical and typographical errors were identified and suggestions for improvement were provided.
Response 6: Thank you for your advice. Fix the following errors.
